# Epidemiology and molecular characterization of Feline panleukopenia virus from suspected domestic cats in selected Bangladesh regions

**Ajran Kabir**[1], **Tasmia Habib**[1], **Chandra Shaker Chouhan**[2], **Jayedul Hassan**[1], **A. K. M. Anisur Rahman**[2], **K. H. M. Nazmul Hussain Nazir**[1,3]*

**1** Department of Microbiology and Hygiene, Bangladesh Agricultural University, Mymensingh, Bangladesh, **2** Department of Medicine, Bangladesh Agricultural University, Mymensingh, Bangladesh, **3** Mymensingh Pet Clinic and Research Center, Mymensingh, Bangladesh

* nazir@bau.edu.bd

**Data Availability Statement:** Data are available from NCBI OQ291236-OQ291241.

## Abstract

Feline panleukopenia (FPL) is a highly contagious cat disease and is endemic in Bangladesh. The study aims to describe the epidemiology and molecular characterization of the Feline panleukopenia virus from the suspected domestic cats in selected Bangladesh regions. Randomly, 161 rectal swabs were collected from the pet hospitals between July 2021 and December 2022. A structured questionnaire was administered through face-to-face interviews with cat owners in order to collect data on potential risk factors for FPL, such as age, sex, sharing litter boxes and every day utensils in multicat households, vaccination history, hospital visits for other diseases, and season. The rectal swabs were tested by PCR targeting the VP2 capsid protein gene, and six PCR-positive samples were further sequenced for molecular characterizations. The risk factors for FPLV were identified using multivariable logistic regression analysis. The overall prevalence of FPL among suspects was 22.9%. The mortality and case fatality were 10.6%, and 45.9%, respectively. However, mortality in kittens was significantly higher (16.4%) than younger cats. The odds of FPL were 8.83 times (95% CI: 3.14–24.85) higher among unvaccinated cats than vaccinated cats. The winter season had almost six times (95% CI: 1.38–24.40) higher odds of FPL than rainy season. In a multicat house, the odds of FPL was about five times (95% CI: 1.93–13.45) higher for cats that shared a litter box and food utensils compared to those that did not engage in such sharing. Visiting hospitals for other reasons nearly triples the odds of FPL (OR: 2.80, 95% CI: 1.04–7.54) compared to cats that do not visit hospitals. Analysis of partial sequence of the VP2 gene revealed genetic variations among the isolates from different regions. Among these isolates, four were identical to FPLV isolates from South Korea and China, while one showed complete homology with FPLV isolates from Thailand. In contrast, the remaining one was 100% identical to *Carnivore protoparvovirus-1* isolated from a feline sample in Italy. Our isolates were classified into three distinct clades alongside Feline panleukopenia virus and *Carnivore protoparvovirus-1*. One in every three suspected cats was infected with Feline panleukopenia. Regular vaccination of the cats, especially those that share common litter box and food utensils and visit hospitals for other purposes, will help reduce the prevalence of FPL in Bangladesh. Besides, it is worth emphasizing the

**Funding:** The research work was conducted using personal funds.

**Competing interests:** The authors have declared that no competing interests exist.

existence of genetic diversity among the circulating Feline panleukopenia viruses in Bangladesh.

## Introduction

Feline panleukopenia (FPL) is a highly contagious and fatal viral disease of cats. The clinical signs of FPL include severe depression, vomiting, diarrhea, a sharp drop in white blood cells (WBCs), and damage to intestinal mucosa resulting in enteritis, dehydration, and ultimately death [1, 2]. The FPL virus (FPLV) can infect both domestic and wild felids (suborder Feliformia), as well as a few wild canids (suborder Caniformia) such as raccoons and foxes. This virus does not infect domestic dogs [3, 4]. FPLV belongs to the family *Carnivore protoparvovirus-1*and is a small, nonenveloped, linear, single-stranded DNA virus with a genome of 5.1-kb [5]. The basic structure of the virus involves two non-structural proteins (NS1 and NS2) and two structural capsid proteins (VP1 and VP2) that are encoded by two open reading frames (ORFs) [3, 4].

The FPLV is usually transmitted through the faecal-oral route, and flea bites. The virus is known to persist long time e in the environment for extended periods, such as up to one year in contaminated organic materials [6, 7]. The prevalence of FPL has been reported in various regions worldwide, including Iran [8], Canada [9], East Africa [10], Spain [11], Vietnam [12], Central West Saudi Arabia [13], and Brazil [14] Among infected cats, young kittens under one year of age experience the highest morbidity and mortality rates [15, 16]. The mortality rate for acute panleukopenia ranges from 25–90%, while per acute infections can result in 100% mortality [17]. In multicat households, the risk of contracting the disease is higher for unvaccinated and younger cats [18]. Although unvaccinated young kittens are more susceptible, cats of any age, sex, or breed can be infected [19, 20].

The clinical manifestations serve as reliable indicators for a presumptive diagnosis of FPL. However, for confirmation, the molecular test (PCR) is recommended [21]. Various PCR protocols targeting the amplification of the VP2 gene have been established and reported for the detection of FPLV [22].

Due to widespread vaccination worldwide, the prevalence of the Feline panleukopenia has significantly decreased over the past few decades [23]. However, in Bangladesh, the pet population, especially cats, is on the rise. Unfortunately, due to lack of awareness among the owners regarding vaccination schedules, hygiene practices, and proper husbandry, cats in Bangladesh are at a higher risk of contracting the Feline panleukopenia virus [24].

In addition, there is limited molecular data available in Bangladesh regarding feline panleukopenia. A recently published study reported a 18.37% prevalence of FPL from diarrheic cats in Bangladesh, while a previous study documented a 22.41% prevalence in the Tangail district. So far, only a single isolate has been molecularly characterized showing 99.71% homology with another FPLV isolate from the United Arab Emirates [25–27]. However, there is lack of reported epidemiological studies in Bangladesh that describe the risk factors for FPL in suspected cats, along with molecular characterization. Therefore, the objectives of this study are to identify risk factors for FPL, detect the virus in the rectal swab of FPL-suspected cats visiting pet clinics, and provide molecular characteristics of the VP2 gene of FPLV from selected regions of Bangladesh.

## Materials and methods

### Ethical approval and informed consent

Verbal consent was obtained from the cat owners to collect samples used in this work. The samples were collected following standard procedures for sample collection, ensuring no harm

was caused to the animals. The study received approval from the Animal Welfare and Experimental Operational Guidance Committee of the university (Approved permit number AWEEC/BAU/2021(56)).

## Study and target population and sample size

The study population consisted of cats visiting pet clinics, while our target population comprised domesticated cats in urban Bangladesh were. The sample size for this study was calculated using Eq 1 as given below:

$$n = \frac{1.96^2 PQ}{d^2} \tag{1}$$

In the above equation, 1.96 = Z value for a 95% confidence level, Expected prevalence (10% = 0.10) [28] and Q = 1-P and d = precision = 5% = 0.05.

Based on these assumptions, the calculated sample size was 138. However, a total of 161 samples were collected for this study.

## Sample and data collection

A total of 161 rectal swabs were collected randomly from suspected cats visiting different pet clinics in three districts of Bangladesh (Fig 1). The study was conducted from July 2021 to December 2022. Rectal swabs were collected using a cotton apparatus and placed in 1ml of phosphate buffer saline (PBS) The samples were then stored at 4°C until transported to the Department of Microbiology and Hygiene, Bangladesh Agricultural University [29]. After

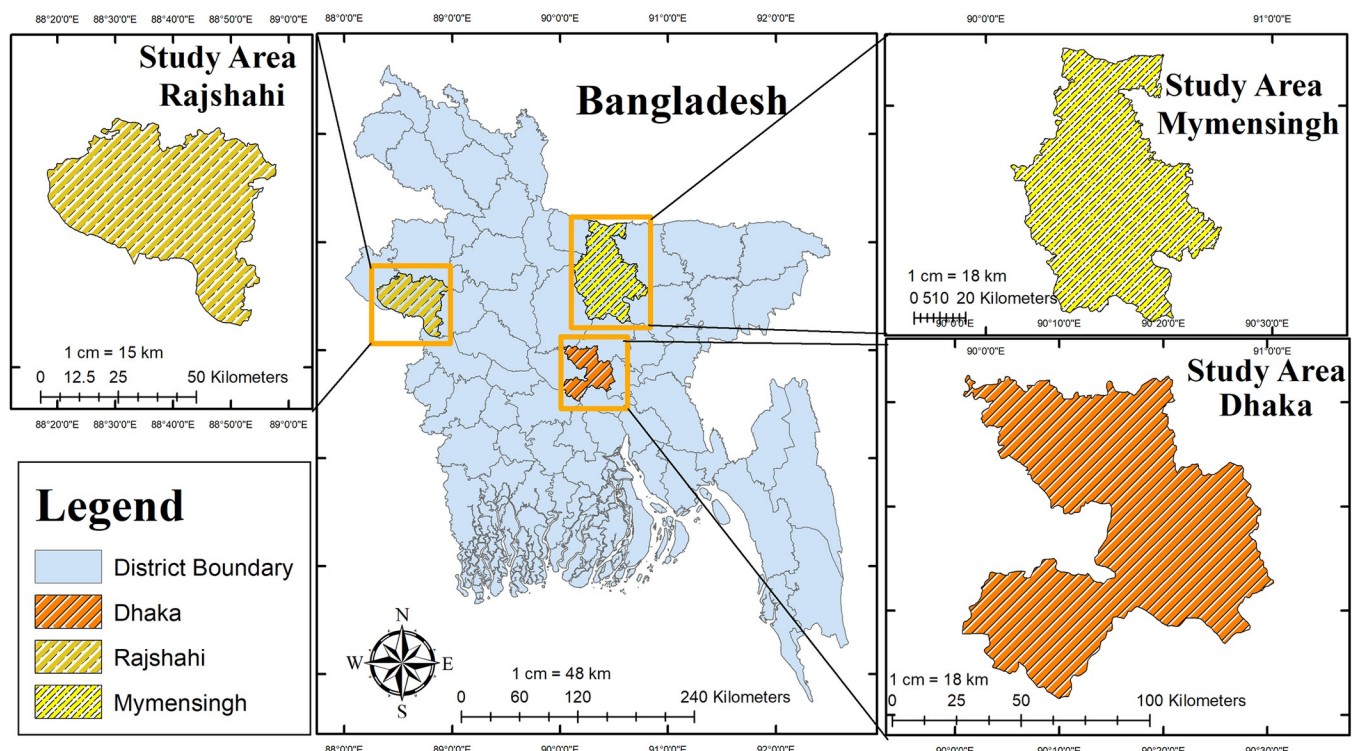

**Fig 1. Bangladesh map showing different sampling areas of Feline panleukopenia virus.** Note: This map is created by our team using ArcGIS version 10.1 (http://www.esri.com/arcgis).

centrifugation at 10,000 RPM for 10 minutes, a 200ul supernatant was collected and stored at -80 ˚C until further processing.

During the sample collection, data regarding possible risk factors for FPL were obtained, through face-to-face interviews with cat owners, using a pre-structured questionnaire. The data collected included details about the cat's age, sex, whether they shared litter boxes and food utensils in multicat households, vaccination history, whether the cat had been taken to hospitals for other diseases, and the specific season. Telephone numbers were also recorded during the interview. Additionally, information on mortality was obtained through telephone conversations with the cat owners and recorded in our questionnaire.

## DNA extraction and PCR

Genomic DNA was extracted from the collected supernatant using AddPrep Viral Nucleic Acid Extraction Kit (AddBio\Korea) following the manufacturer's instructions [19]. The purity of the extracted DNA was assessed using a nanodrop spectrophotometer (Thermo \USA) by measuring the absorbance at a wavelength of 260/280 nm. The DNA purity was determined within the range of 1.8 to 2.1 [30]. The DNA extracted from the vaccine (NOBI-VAC® FELINE 1-HCP) was used as a positive control in the PCR assay.

PCR was performed using specific primer pairs targeting the VP2 gene residue of FPLV. The PCR reaction mixtures, consisting of 25 ul, were prepared by combining FM-F (5'-GCT TTA GAT GAT ACT CAT GT -3') and FM-R (5'-GTA GCT TCA GTA ATA TAG TC-3') primers, as described in a previous study [27]. The reaction mixtures were then put in a thermocycler machine maintaining the following conditions: initial denaturation at 95˚C for 5 min, 35 cycles of denaturation at 95˚C for 30 s, annealing at 60˚C for 1 min, extension at 72˚C for 1 min, and a final extension at 72˚C for 10 minutes. Subsequently, the PCR-amplified products were separated on a 1.5% agarose gel in Tris-acetate EDTA buffer and visualized using an ultraviolet transilluminator (Syngene).

## FPLV VP2 gene partial sequencing

Six PCR-positive amplicons of the VP2 gene were selected, with two from each location. They were then purified using the FavorPrep™ GEL/PCR Purification Kit (Biotech Corp/ USA) according to the manufacturer's instruction. The purified amplicons were subjected to Sanger' sequencing using the Applied BioSystems 3130 automated DNA sequencer. The resulting chromatograms were visualized and trimmed using Chromas 2.6.6 software. To confirm the sequence homology a Basic Local Alignment Search Tool (BLAST) analysis was performed in the NCBI database (BLAST: Basic Local Alignment Search Tool (nih.gov)). VP2 gene sequences with the highest coverage (100%) and highest identities (99%-100%) to our sequences from different parts of the world were retrieved from the NCBI nucleotide database. Multiple sequence alignment was performed using the Clustal W algorithm in MEGA11 [31]. The nucleotide sequences were submitted to the NCBI Nucleotide Database GenBank using BankIt https://submit.ncbi.nlm.nih.Gov/subs/ GenBank. The FPV gene fragment sequences were deposited in the GenBank under accession numbers OQ291236-OQ291241.

## Phylogenetic analysis

The obtained sequences from this study, along with 27 other parvovirus reference sequences obtained from GenBank (NCBI) were aligned. The evolutionary history was inferred using the neighbor-joining method based on the Tamura-Nei model, implemented in MEGA11 [31]. The branch lengths in the phylogenetic tree were measured in terms of the number of

substitutions per site. A bootstrap test (1000 replicates) was conducted to evaluate the confidence level of branching in the phylogenetic tree.

## Statistical analysis

The Microsoft Excel 2019 spreadsheet was used for data entry and management, and the data was then transferred into SPSS 22 for analysis [32]. The survey responses were categorized, and continuous variables were transformed into categorical forms (age, sex, sharing litter box and food utensils in multicat households, vaccination history, visit to the hospital for other purposes, season, etc.). The prevalence, mortality, and case fatality variations among districts and age groups were assessed using Z-tests for proportions. Initially, the association between Feline Panleukopenia virus infection and different explanatory variables was evaluated using univariable logistic regression models. Explanatory variables with a p-value $\leq 0.2$ were selected for inclusion in the multivariable logistic regression analysis. Prior to conducting the multivariable logistic regression analysis, the variation inflation factor (VIF) was used to check for multicollinearity among the explanatory variables with a threshold of $\leq 5$. A backward model selection strategy was employed for the final model selection. The Hosmer-Lemeshow test was used to evaluate the overall model's fit.

## Results

### Prevalence, mortality, and case fatality

Out of 161 suspected cats, 22.9% tested positive for FPLV in PCR. The overall mortality and case fatality were 10.6% and 45.9%, respectively (Table 1).

The highest prevalence was observed in Dhaka district (26.3%) and lowest in Rajshahi district (18.4%). On the other hand, Mymensingh district's mortality (16.4%) and case fatality (69.2%) were the highest. The prevalence (29.4%) and case fatality (55.0%) were higher among kittens aged 0 - $\leq 6$ months. However, but the mortality (16.4%) was significantly higher in kittens (0 - $\leq 6$ months) than young cats (7–24 months) (Table 1).

### Risk factors of FPL in suspected cats

Five variables were found to be associated with FPL at p-values $\leq 0.2$ in the univariable screening and were included in the multivariable model (Table 2). However, in the final multivariable logistic regression model, only four variables showed significant associations with FPL among the suspected cats (Table 3). The absence of a vaccination history, sharing litter boxes and food utensils, visiting hospitals for other purposes, and the winter seasons were all significantly associated with FPL. Cats without a vaccinating history had 8.83 times higher odds of FPL compared to vaccinated cats, with a 95% confidence interval (CI) of 3.14–24.385. The

**Table 1. Age and district-wise prevalence, mortality, and case fatality of Feline panleukopenia virus infection.**

| Variable | Category | Tested | Positive | Died | Prevalence (%) | Mortality (%) | Case fatality (%) |
|---|---|---|---|---|---|---|---|
| District | Dhaka | 57 | 15 | 5 | 26.3[a] | 8.8[a] | 33.3[a] |
| | Mymensingh | 55 | 13 | 9 | 23.6[a] | 16.4[a] | 69.2[a] |
| | Rajshahi | 49 | 9 | 3 | 18.4[a] | 6.1[a] | 33.3[a] |
| Age (months) | Kitten (0-$\leq 6$) | 67 | 20 | 11 | 29.4[a] | 16.4[a] | 55.0[a] |
| | Young (7–24) | 94 | 17 | 6 | 18.1[a] | 6.4[b] | 35.3[a] |
| Overall | | 161 | 37 | 17 | 22.9 | 10.6 | 45.9 |

[ab] values with different superscripts within the same column for each variable differ significantly (p < 0.05).

**Table 2. Univariable association between Feline panleukopenia virus infection with explanatory variables in selected districts of Bangladesh.**

| Risk factors | Category level | Tested | Prevalence (%) | OR | 95% CI | P-Value |
|---|---|---|---|---|---|---|
| Age | Kitten (0- ≤ 6 month) | | | 1.93 | 0.92–4.05 | .083 |
| | Young (7–24 month) | | | Ref | - | - |
| Sex | Male | | | Ref | - | - |
| | Female | | | 1.52 | 0.72–3.19 | .274 |
| Vaccination history | Yes | | | Ref | - | - |
| | No | | | 6.48 | 2.52–16.65 | < .001 |
| Sharing litter box and food utensils in multicat households | No | | | Ref | - | - |
| | Yes | | | 4.01 | 1.81–8.86 | .001 |
| Visit to the hospital for other purposes | No | | | Ref | - | - |
| | Yes | | | 3.17 | 1.48–6.79 | .003 |
| Season | Summer (Mar-Jun) | | | 1.82 | 0.45–7.40 | .405 |
| | Rainy (Jul-Oct) | | | Ref | - | - |
| | Winter (Nov-Feb) | | | 3.04 | 0.84–11.04 | .091 |

OR: Odds ratio; CI: 95% confidence interval

winter season had almost six times higher odds of FPL (95% CI: 1.38–24.40) compared to rainy season. In multicat households, cats that shared a litter box and food utensils had about five times (95% CI: 1.93–13.45) higher odds of FPL in compared to those that did not share. Furthermore, visiting hospitals for other reasons nearly triples the odds of FPL (OR: 2.80, 95% CI: 1.04–7.54) compared to cats who do not visit hospitals.

## BLAST analysis and sequence alignment

The VP2 capsid protein gene was targeted for the molecular detection of FPLV. All the positive samples showed sensitivity to the specific primers, resulting in a 698 bp amplicon size under UV (Fig 2). For further molecular characterization, six representative isolates from different locations were selected and sequenced, as, listed in Table 4.

Upon conducting BLAST analysis, it was determined that our isolates THBAU3 (OQ291238), THBAU4 (OQ291239), THBAU5 (OQ291240), THBAU6 (OQ291241) exhibited 100% homology with previously confirmed Feline panleukopenia isolates of South Korea (MN400978.1) and China (MF541135.1). In addition, isolate THBAU2 displayed 100%

**Table 3. The results of the final multivariable logistic regression model to identify risk factors for Feline panleukopenia virus infection in selected Bangladesh districts.**

| Risk factors | Category level | OR | 95% CI | P-Value |
|---|---|---|---|---|
| Vaccination history | Yes | Ref | - | - |
| | No | 8.83 | 3.14–24.85 | < .001 |
| Sharing litter box and food utensils in multicat households | No | Ref | - | - |
| | Yes | 5.09 | 1.93–13.45 | .001 |
| Visit to the hospital for other purposes | No | Ref | - | - |
| | Yes | 2.80 | 1.04–7.54 | .042 |
| Season | Summer (Mar-Jun) | 2.18 | 0.48–10.01 | 0.314 |
| | Rainy (Jul-Oct) | Ref | - | - |
| | Winter (Nov-Feb) | 5.81 | 1.38–24.40 | .016 |

OR: Odds ratio; CI: 95% confidence interval

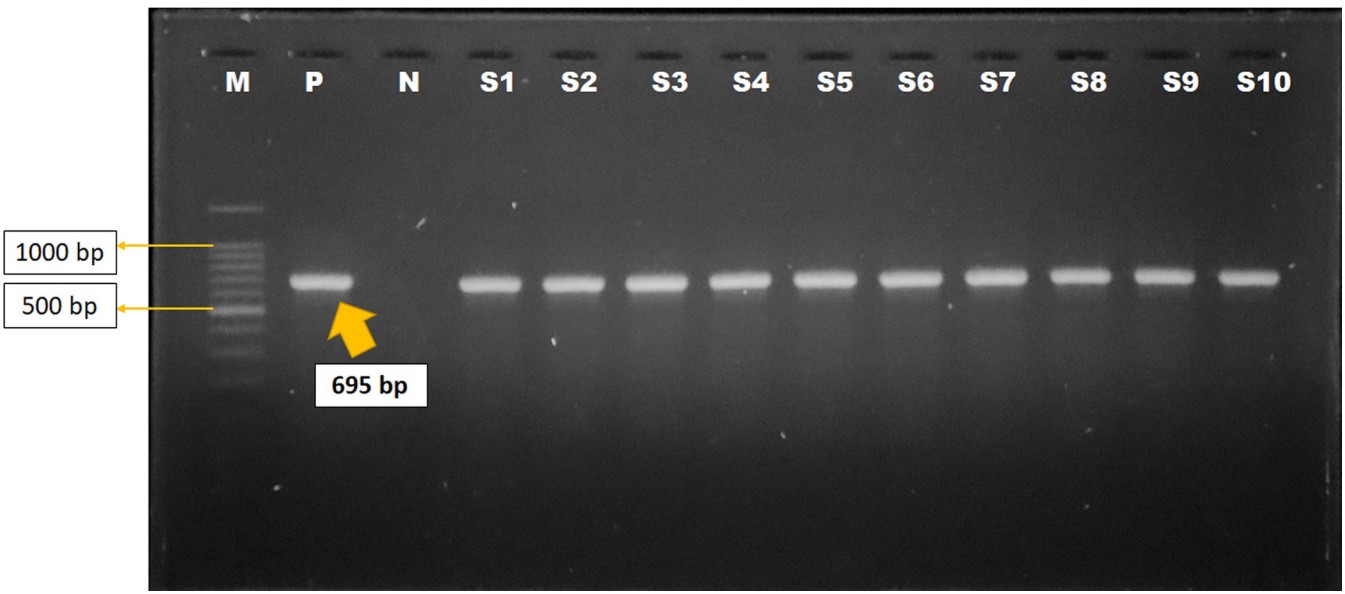

**Fig 2. Identification of Feline panleukopenia virus by polymerase chain reaction.** Gel electrophoresis showing VP2 capsid protein genes amplicons of Feline Panleukopenia virus (695bp). In Lanes: M- 100 bp DNA ladder (Promega, USA) P: Positive control N: Negative control, Lane S1-S10: Complies with the Samples of FPLV showing approximately 695 bp.

identity to a Feline panleukopenia isolate from Thailand (MN127779.1), while THBAU1 showed 100% homology with *Carnivore protoparvovirus-1* (MW847199.1) isolated from a feline sample in Italy.

## Phylogenetic analysis

For phylogenetic analysis, the VP2 gene sequences of Feline Panleukopenia, *Carnivore protoparvovirus-1*, Canine Parvovirus 2 (CPV-2), Mink enteritis virus (MEV), and Raccoon parvovirus (RaPV) were retrieved from the NCBI database.

The phylogenetic analysis revealed the presence of five distinct clades. Among our six isolates, THBAU1 clustered with *Carnivore protoparvovirus-1* isolates, while the remaining isolates formed two separate clades along with Feline Panleukopenia virus isolates (Fig 3).

## Discussion

One in every three FPL suspect cats was found to be infected with FPLV. Non-vaccination, sharing of litter boxes and food utensils, seasonal variation, and visiting pet hospitals were identified as risk factors for FPL. We recommended regular vaccination for cats, especially

**Table 4. The list of sequenced isolates with their accession number and location.**

| No | Isolates | Accession Number | Organism | Location |
|---|---|---|---|---|
| 1 | THBAU1 | OQ291236 | *Carnivore protoparvovirus 1* | Mymensingh |
| 2 | THBAU2 | OQ291237 | Feline panleukopenia virus | Mymensingh |
| 3 | THBAU3 | OQ291238 | Feline panleukopenia virus | Dhaka |
| 4 | THBAU4 | OQ291239 | Feline panleukopenia virus | Dhaka |
| 5 | THBAU5 | OQ291240 | Feline panleukopenia virus | Rajshahi |
| 6 | THBAU6 | OQ291241 | Feline panleukopenia virus | Rajshahi |

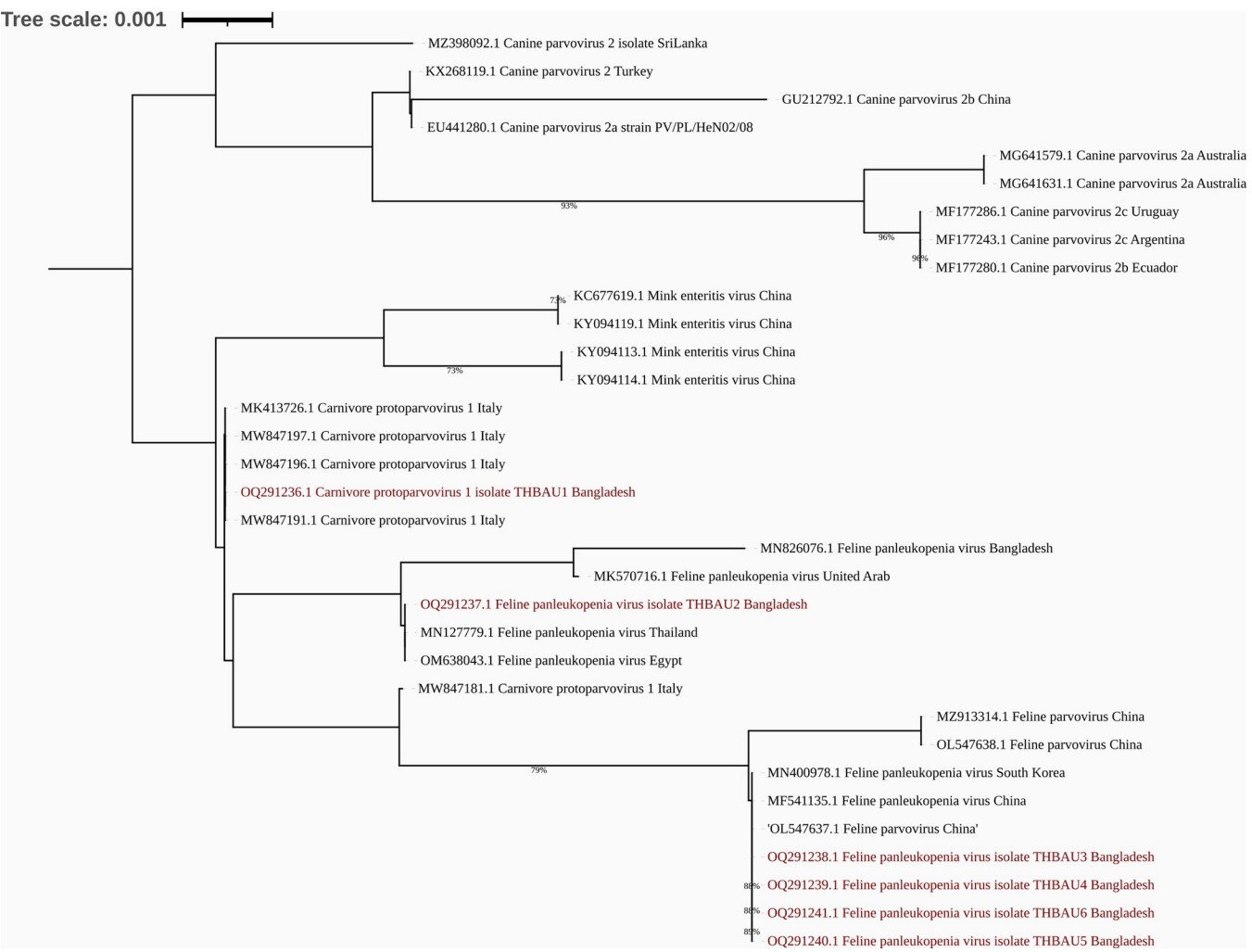

**Fig 3. Phylogenetic tree of FPLV.** The Neighbor-Joining tree was constructed using 33 partial VP2 gene sequences of *Carnivore protoparvovirus-1*, Feline panleukopenia virus, Canine parvovirus, Mink enteritis virus, and Raccoon parvovirus. The evolutionary distances between sequences were calculated using the Tamura-Nei method. The evolutionary analyses were conducted using MEGA11 software [31] and the tree was visualized in iTOL. In the tree, the sequences obtained in this study are highlighted in red.

those that share litter box and food utensils, and those that visit hospitals to reduce the prevalence of FPL in the Bangladesh context.

Pet animal rearing, particularly cats, is gaining popularity in Bangladesh due to their ease of care. However, the increasing prevalence of FPL, a highly contagious and deadly disease, poses a significant threat to the cat population. Unfortunately, many cat owners in Bangladesh are unaware of the disease and lack proper management practices. This study represents the first comprehensive investigation into the epidemiology and molecular characteristics of FPL in Bangladesh.

The prevalence of FPL observed in our study was 23%, which is higher than the prevalence reported in a previous study [27]. However, our findings align with the results reported in studies [25, 26]. The relatively higher prevalence observed in our study may be attributed to the sampling method employed. It is important to note that our study specifically collected samples from suspected domestic cats visiting pet hospitals, rather than randomly sampling from the general cat population. This targeted sampling approach could have influenced the

prevalence and may not be representative of the overall cat population in Bangladesh. This study also revealed a mortality of 10.6% with a case fatality of 45.5%. Several factors contribute to these high figures, including a lack of understanding among pet owners regarding proper pet management, delayed veterinary visits, inadequate vaccination, and the absence of a specialized pet hospital in our country. In addition, this study exhibited a significant correlation between FPL and mortality in Kitten aged 0-≤ 6 months compared to young cats aged 7–24 months. This might be because most kittens were unvaccinated and had lower immunity levels [15, 16].

The pet owners who administered the FPLV vaccine to their cats significantly reduced the occurrence of FPLV compared to those who did not, which is consistent with findings reported in [21]. This is possible as vaccinated cats substantial quantities of antibodies that provide protection against FPLV infection. In addition, vaccinated cats that contract FPLV exhibit milder clinical signs and symptoms compared to unvaccinated cats with FPLV-infection [33].

Our study also revealed a strong association between sharing litter box and food utensils in multicat households and the occurrence of FPL. The practice of using the same litter box and eating utensils can facilitate the transmission of the disease, as healthy cats can come into contact with the virus through their saliva, urine, feces, nasal secretions, tears, and contaminated clothing [18, 34].

Intriguingly, visiting a hospital for other purposes with cat significantly increases the odds of FPLV infection. Owners may interact with others' cats while receiving treatment in the hospital, and the same thermometer and premises are used for both sick and healthy cats. The premises are not always disinfected immediately after treating a sick animal, and it is possible for the same doctor to examine both sick and healthy cats. As a result, many cats brought to hospitals for other issues become infected with diseases from diseased animals.

Hospital management can play a crucial role in preventing nosocomial FPLV infections by implementing effective infection control policies and procedures. This includes regular cleaning and disinfection of areas where animals come into contact, including potentially contaminated accessories. Moreover, staff and visitors entering pet wards must practice good personal hygiene such as hand washing before interacting with animals. Evidence of such nosocomial infections has been reported in previous studies [35, 36].

The seasonal pattern was also a significant factor in the occurrence of FPL. Our investigation observed a strong association between the winter season and occurrence of FPL. FPL viruses have a higher likelihood of infecting cats in colder and drier climates, and the cold weather during the winter season can affect the immune response of cats. Additionally, seasonal weight loss, which is also observed during the winter season, may contribute to the occurrence of disease [37].

The VP2 protein serves as the major capsid protein for *Carnivore protoparvovirus-1* and plays a crucial role in determining the host range of the virus It is also subject to antibody-mediated selection [38]. Mutations in specific amino acids of the VP2 capsid protein are responsible for the variation observed among different strains of parvovirus [27]. In our study, four out of six isolates (THBAU3, THBAU4, THBAU5, THBAU6) that were partially sequenced showed 100% homology with previously published isolates from South Korea (MN400978.1, MW035309.1, HQ184200.1) and China (MF541135.1, OM918773.1, OL547637.1, MT178243.1). These isolates were obtained from the Dhaka and Rajshahi regions of Bangladesh. On the other hand, our isolate THBAU2 displayed 100% homology with FPLV isolate18R217C/TH/2018 from Thailand. Moreover, our remaining isolate THBAU1 demonstrated 100% homology with *Carnivore protoparvovirus 1* isolate 44350 from Italy.

A previous study conducted in Bangladesh, focusing on the molecular characterization of the VP2 gene of feline panleukopenia virus, revealed a high level of homology with our isolates THBAU1 (more than 99.52%) and THBAU2 (99.62%). However, there is less than 99% homology observed with the other sequences analyzed. When compared to the e previously published sequence of Bangladesh [27]. Some point mutations were identified in the nucleotide sequences of our isolates'. Notably, no amino acid mutations were observed. Based on these findings, it can be concluded that several mutated variants of the virus are circulating in different regions of Bangladesh.

In our phylogenetic analysis, we observed that four isolates (THBAU3, THBAU4, THBAU5, and THBAU6) clustered together with previously identified Feline Panleukopenia viruses from South Korea and China. However, isolate THBAU2 formed a separate cluster along with Feline Panleukopenia viruses identified in Bangladesh, Egypt, and UAE. Furthermore, isolate THBAU1 formed a distinct cluster with published *Carnivore protoparvovirus-1* isolates. Therefore, it can be assumed that these isolates may have been introduced to Bangladesh from the mentioned countries.

In this study, we conducted partial sequencing of the VP2 gene, which provides limited information regarding the antigenic and genetic differences between the original parvovirus and its variants. However, to gain a comprehensive understanding, a whole genome or full-length VP2 gene sequence is required. Due to lack of fund and laboratory facilities, we were only able to perform partial sequencing. The study also had limitations in terms of virus culture and its pathogenicity observation. Additionally, investigating predictive laboratory variables, such as blood profiles, between survivors and non-survivors would contribute to a more comprehensive investigation of Feline Panleukopenia in Bangladesh.

## Conclusion

One in every three suspected cats was found to be infected with feline panleukopenia, and the disease exhibited a high case fatality, particularly among young cats and in a specific region. Regular vaccination of the cats, especially those that share a common litter box, food utensils, and those that visit hospitals for other purposes, will help reduce the prevalence of FPL in Bangladesh. Moreover, the genetic variation observed in this study provides valuable insights into the evolution of FPLV in Bangladesh. This information will aid in the development of effective prevention and control strategies for combating this deadly disease.

## Supporting information

**S1 Raw images.**
(PDF)

## Acknowledgments

The authors would like to thank Dr. Sagir's Pet Clinic & Research Center, Mymensingh Pet Clinic and Research Center, Veterinary Teaching Hospital, Bangladesh Agricultural University, and Birds and Pet Animal Clinic, Rajshahi, for their support during sample collection.

## Author Contributions

**Conceptualization:** Ajran Kabir, Tasmia Habib.

**Data curation:** Ajran Kabir, Tasmia Habib, Chandra Shaker Chouhan.

**Formal analysis:** A. K. M. Anisur Rahman.

**Funding acquisition:** K. H. M. Nazmul Hussain Nazir.

**Methodology:** Ajran Kabir.

**Supervision:** Jayedul Hassan, K. H. M. Nazmul Hussain Nazir.

**Writing – original draft:** Ajran Kabir, Chandra Shaker Chouhan.

**Writing – review & editing:** Jayedul Hassan, A. K. M. Anisur Rahman, K. H. M. Nazmul Hussain Nazir.

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
