## [Decision Letter · Decision Letter 0]

17 Apr 2023

PONE-D-23-04863Epidemiology and molecular characterization of Feline Panleukopenia Virus from suspected domestic cats in BangladeshPLOS ONE

Dear Dr. Nazir,

Thank you for submitting your manuscript to PLOS ONE. After careful consideration, we feel that it has merit but does not fully meet PLOS ONE’s publication criteria as it currently stands. Therefore, we invite you to submit a revised version of the manuscript that addresses the points raised during the review process.

We look forward to receiving your revised manuscript.

Kind regards,

Sherin Reda Rouby, PhD

Academic Editor

PLOS ONE

“The research work was conducted using personal funds.”

Reviewers' comments:

Reviewer's Responses to Questions

**Comments to the Author**

1. Is the manuscript technically sound, and do the data support the conclusions?

Reviewer #1: Partly

Reviewer #2: Yes

Reviewer #3: Partly

2. Has the statistical analysis been performed appropriately and rigorously? 

Reviewer #1: Yes

Reviewer #2: I Don't Know

Reviewer #3: No

3. Have the authors made all data underlying the findings in their manuscript fully available?

Reviewer #1: Yes

Reviewer #2: Yes

Reviewer #3: Yes

4. Is the manuscript presented in an intelligible fashion and written in standard English?

Reviewer #1: Yes

Reviewer #2: Yes

Reviewer #3: No

5. Review Comments to the Author

Reviewer #1: In this manuscript, the authors investigated the epidemiology of FPLV in Bangladesh and gave a phylogenetic analysis of FPLV VP2 gene. However, initial assessment indicated that this study lacks novelty and I have some major concerns and suggestions that need to be addressed.

1. In my opinion, as an epidemiological investigation and analysis, the number of samples collected in this study is not enough, and the samples were only collected from the pet hospitals. It is preferable to be able to collect samples from stray cats as well as cat breeding farms for testing to make the analysis more comprehensive.

2. The authors detected the FPLV from the hospital samples, furthermore, the prevalence of other pathogens and the co-infections with FPLV would be better mentioned.

3. It would be better to sequence the complete sequence of VP2 gene.

4. The authors detected samples from the suspected cat for FPLV, please provide more information on subsequent treatment and prognosis of the Feline Panleukopenia.

Reviewer #2: Overall I think this is a well written manuscript about a common disease in a location where not much is known about prevalence. I think that the authors are missing an opportunity to discuss in greater detail what I see as the most significant finding; that this has a significant link to veterinary visits. I would like to see a more expanded discussion about the need for increased biosecurity in regards to veterinary visits. Otherwise, well done.

Reviewer #3: The authors explore" Epidemiology and molecular characterization of Feline Panleukopenia Virus from suspected domestic cats in Bangladesh". However, the study was needed to some modifications as following to be ready for publications. After reading I have some concerns itemized below:

1-In introduction section: please rewrite the family of FLP (carnivore protoparvovirus-1) in italic form.

2-Did the author take rectal swab samples or fecal samples? Please clarify this information throughout the entire manuscript.

3- please, added the fig 1,……….and so on without title in the text, but the remaining details added in the right site.

4- Please, remove the last row in the table 1.

5- Please, added the word (month) near to 6, 7, 24 in the table 2.

6-Is the virus isolate (THBAU2) identical to FLP isolate from Egypt or Thailand?

7- Is the ladder 100kbp or 1kbp? From which company was it made?

8- Complete sequencing of VP2 gene is required In order to stand for certain results.

9-Virus isolation is important for diagnosis of FPL virus infection as well as fundamental studies of FPL, so, the authors will need to isolate the virus from cats.

10-The all manuscript is needed to grammar and spelling checker. Please check the entire manuscript for grammatical errors& I advise the revising for English Language by a native English speaker.

11-The statistical evaluation of the approach is little bit low.

12-Discussion part is very weak and has a great lack of evidence and scientific explanation and can’t be accepted in that way. Interpretate your results according to the evidence based knowledge and discuss it more.

13- Write your conclusion clearly after the discussion part.

14- The PCR figure is suspicious.

15- The phylogenetic analysis figure is hesitated.

With my best wishes

Fatma Abdallah

6. PLOS authors have the option to publish the peer review history of their article (what does this mean?). If published, this will include your full peer review and any attached files.

Reviewer #1: No

Reviewer #2: **Yes: **Michael Nappier

Reviewer #3: No

---

## [Author Response · Author response to Decision Letter 0]

31 May 2023

The authors are highly obliged and thankful to the reviewer for his constructive comments, which helped us improve the quality of the paper in terms of science and presentation. The weak points were taken care of, and necessary modifications were made and marked with track changes (please see the revised manuscript with track changes).

Reviewer #1: In this manuscript, the authors investigated the epidemiology of FPLV in Bangladesh and gave a phylogenetic analysis of FPLV VP2 gene. However, the initial assessment indicated that this study lacks novelty, and I have some major concerns and suggestions that need to be addressed.

1. In my opinion, as an epidemiological investigation and analysis, the number of samples collected in this study is not enough, and the samples were only collected from the pet hospitals. It is preferable to be able to collect samples from stray cats as well as cat breeding farms for testing to make the analysis more comprehensive.

Response: We appreciate your suggestions for collecting samples from stray cats and cat breeding farms. While our main purpose of this study was on identifying risk factors for FPL in suspected domestic cats, we acknowledge that increasing our sample size to include animals from different settings could provide greater insights. However, due to budget constraints, we were only able to test 161 samples using PCR for FLP diagnosis. Despite the limited sample size, we believe our findings contribute to the understanding of this disease in domestic cats.

2. The authors detected the FPLV from the hospital samples, furthermore, the prevalence of other pathogens and the co-infections with FPLV would be better mentioned.

Response: We did not use multiplex PCR and it was not possible to detect other pathogens. In a future study we can explore that.

3. It would be better to sequence the complete sequence of VP2 gene.

Response: Due to lack of fund, full length VP2 gene sequence was not possible at this time. 

4. The authors detected samples from the suspected cat for FPLV, please provide more information on subsequent treatment and prognosis of the Feline Panleukopenia.

Response: The age and district wise mortality and case fatality were presented in Table 1. 

Reviewer #2: Overall I think this is a well written manuscript about a common disease in a location where not much is known about prevalence. I think that the authors are missing an opportunity to discuss in greater detail what I see as the most significant finding; that this has a significant link to veterinary visits. I would like to see a more expanded discussion about the need for increased biosecurity in regards to veterinary visits. Otherwise, well done.

We greatly appreciate your positive attitude and acknowledgement of the significance of our study. Moreover, we have taken into consideration your valuable feedback and incorporated necessary modifications throughout the manuscript to address the limitations highlighted by you. Specifically, we have included an extensive discussion on the topic of biosecurity pertaining to veterinary visits, aligning with your observations.

Reviewer #3: The authors explore" Epidemiology and molecular characterization of Feline Panleukopenia Virus from suspected domestic cats in Bangladesh". However, the study was needed to some modifications as following to be ready for publications. After reading I have some concerns itemized below:

1-In introduction section: please rewrite the family of FLP (carnivore protoparvovirus-1) in italic form.

Response: Changed according to comments, Carnivore protoparvovirus-1

2-Did the author take rectal swab samples or fecal samples? Please clarify this information throughout the entire manuscript.

Response: Thank you for this comment. We used rectal swab samples, and changed throughout the entire manuscript.

3- please, added the fig 1 ,……….and so on without title in the text, but the remaining details added in the right site.

Response: Added as per your direction, This map is created by our team using ArcGIS version 10.1 (http://www.esri.com/arcgis)

4- Please, remove the last row in the table 1.

Response: Removed according to your comment

5- Please, added the word (month) near to 6, 7, 24 in the table 2.

Response: Added

6-Is the virus isolate (THBAU2) identical to FLP isolate from Egypt or Thailand?

Response: The virus isolate (THBAU2) is identical to FLP isolate from Thailand

7- Is the ladder 100kbp or 1kbp? From which company was it made?

Response: In Lanes: M- 100 bp DNA ladder (Promega, USA)

8- Complete sequencing of VP2 gene is required In order to stand for certain results.

Response: Due to lack of fund, full length VP2 gene sequence was not possible. We modified our results according to our findings. Please go through the revised manuscript (Revised Manuscript with Track Changes) for more details

9-Virus isolation is important for diagnosis of FPL virus infection as well as fundamental studies of FPL, so, the authors will need to isolate the virus from cats.

Response: Virus isolation was not possible due to inadequate lab facilities.

10-The all manuscript is needed to grammar and spelling checker. Please check the entire manuscript for grammatical errors& I advise the revising for English Language by a native English speaker.

Response: We have made an extensive English edit.

11-The statistical evaluation of the approach is little bit low.

Response: We apologize for not being to understand this comment. Please make specific comment where to improve out statistical analysis.

12-Discussion part is very weak and has a great lack of evidence and scientific explanation and can’t be accepted in that way. Interpretate your results according to the evidence-based knowledge and discuss it more.

Response: Discussion part has been modified. Please go through the revised manuscript (Revised Manuscript with Track Changes) for more details.

13- Write your conclusion clearly after the discussion part.

Response: We have added conclusion after the discussion section.

14- The PCR figure is suspicious.

Response: New PCR figure has been added.

15- The phylogenetic analysis figure is hesitated.

Response: Modification has been made in the phylogenetic analysis figure.

---

## [Decision Letter · Decision Letter 1]

7 Jun 2023

Epidemiology and molecular characterization of Feline Panleukopenia Virus from suspected domestic cats in selected Bangladesh regions

PONE-D-23-04863R1

Dear Dr. Nazir,

We’re pleased to inform you that your manuscript has been judged scientifically suitable for publication and will be formally accepted for publication once it meets all outstanding technical requirements.

Kind regards,

Sherin Reda Rouby, PhD

Academic Editor

PLOS ONE

Additional Editor Comments (optional):

Reviewers' comments:

Reviewer's Responses to Questions

**Comments to the Author**

1. If the authors have adequately addressed your comments raised in a previous round of review and you feel that this manuscript is now acceptable for publication, you may indicate that here to bypass the “Comments to the Author” section, enter your conflict of interest statement in the “Confidential to Editor” section, and submit your "Accept" recommendation.

Reviewer #3: All comments have been addressed

2. Is the manuscript technically sound, and do the data support the conclusions?

Reviewer #3: Yes

3. Has the statistical analysis been performed appropriately and rigorously? 

Reviewer #3: Yes

4. Have the authors made all data underlying the findings in their manuscript fully available?

Reviewer #3: Yes

5. Is the manuscript presented in an intelligible fashion and written in standard English?

Reviewer #3: Yes

6. Review Comments to the Author

Reviewer #3: The modifications made by the authors made the research paper ready for international publication, and this is from my professional point of view.

7. PLOS authors have the option to publish the peer review history of their article (what does this mean?). If published, this will include your full peer review and any attached files.

Reviewer #3: No

---

## [Editor Report · Acceptance letter]

13 Jun 2023

PONE-D-23-04863R1 

Epidemiology and molecular characterization of Feline Panleukopenia Virus from suspected domestic cats in selected Bangladesh regions 

Dear Dr. Nazir:

I'm pleased to inform you that your manuscript has been deemed suitable for publication in PLOS ONE. Congratulations! Your manuscript is now with our production department. 

Kind regards, 

on behalf of

Professor Sherin Reda Rouby 

Academic Editor

PLOS ONE